# Xylo-Oligosaccharides in Prevention of Hepatic Steatosis and Adipose Tissue Inflammation: Associating Taxonomic and Metabolomic Patterns in Fecal Microbiomes with Biclustering

**DOI:** 10.3390/ijerph18084049

**Published:** 2021-04-12

**Authors:** Jukka Hintikka, Sanna Lensu, Elina Mäkinen, Sira Karvinen, Marjaana Honkanen, Jere Lindén, Tim Garrels, Satu Pekkala, Leo Lahti

**Affiliations:** 1Faculty of Sport and Health Sciences, University of Jyväskylä, FI-40014 Jyväskylä, Finland; sanna.t.k.lensu@jyu.fi (S.L.); elina.e.makinen@jyu.fi (E.M.); sira.karvinen@jyu.fi (S.K.); marjaana.t.hakala@student.jyu.fi (M.H.); satu.p.pekkala@jyu.fi (S.P.); 2Department of Psychology, University of Jyväskylä, FI-40014 Jyväskylä, Finland; 3Veterinary Pathology and Parasitology and Finnish Centre for Laboratory Animal Pathology/HiLIFE, University of Helsinki, FIN-00014 Helsinki, Finland; jere.linden@helsinki.fi; 4Department of Computing, University of Turku, FI-20014 Turku, Finland; tim.garrels@student.hpi.uni-potsdam.de (T.G.); leo.lahti@utu.fi (L.L.); 5Department of Clinical Microbiology, Turku University Hospital, FI-20521 Turku, Finland

**Keywords:** non-alcoholic fatty liver disease, xylo-oligosaccharides, metabolites, gut microbiota, biclustering, high fat diet, microRNA, rats

## Abstract

We have shown that prebiotic xylo-oligosaccharides (XOS) increased beneficial gut microbiota (GM) and prevented high fat diet-induced hepatic steatosis, but the mechanisms associated with these effects are not clear. We studied whether XOS affects adipose tissue inflammation and insulin signaling, and whether the GM and fecal metabolome explain associated patterns. XOS was supplemented or not with high (HFD) or low (LFD) fat diet for 12 weeks in male Wistar rats (*n* = 10/group). Previously analyzed GM and fecal metabolites were biclustered to reduce data dimensionality and identify interpretable groups of co-occurring genera and metabolites. Based on our findings, biclustering provides a useful algorithmic method for capturing such joint signatures. On the HFD, XOS-supplemented rats showed lower number of adipose tissue crown-like structures, increased phosphorylation of AKT in liver and adipose tissue as well as lower expression of hepatic miRNAs. XOS-supplemented rats had more fecal glycine and less hypoxanthine, isovalerate, branched chain amino acids and aromatic amino acids. Several bacterial genera were associated with the metabolic signatures. In conclusion, the beneficial effects of XOS on hepatic steatosis involved decreased adipose tissue inflammation and likely improved insulin signaling, which were further associated with fecal metabolites and GM.

## 1. Introduction

Up to 90% of the obese population in Western countries is estimated to suffer from non-alcoholic fatty liver disease (NAFLD). NAFLD is defined as excessive fat accumulation in the liver, which is not caused by excessive alcohol consumption or steatogenic drug use. Without intervention, simple NAFLD can progress to steatohepatitis, which is characterized by steatosis along with inflammation and hepatocyte degeneration, and ultimately cirrhosis, exposing the patient to a risk of hepatic failure and hepatocellular carcinoma [1].

In recent years, the pathogenesis of liver diseases has been shown to involve the digestive system and in particular the microbes that it hosts [2]. A very likely explanation could be the flux of microbial metabolites into the portal circulation, facilitated by inflammation in the gastrointestinal tract [3]. Understanding the role of the gut microbiota (GM) in NAFLD has raised the hope that the GM could be modulated to alleviate the disease with, for instance, specific diets. We have shown that high hepatic fat content was associated with low abundance of *Faecalibacterium prausnitzii* in humans [4,5] and, further, that the intragastric administration of *F. prausnitzii* ameliorated NAFLD in mice [6]. We and others have found that the growth of *F. prausnitzii* can be naturally increased by feeding it with prebiotic xylo-oligosaccharides (XOS) [5,7] and we further found that XOS supplementation partly prevented hepatic steatosis and increased *F. prausnitzii* abundance in rats [5]. However, besides hepatic metabolism, serological findings and gut integrity, the mechanisms associated with prebiotic-induced prevention of NAFLD have not been widely explored. Hepatic fat accumulation has been shown to strongly associate with adipose tissue insulin resistance [8] and inflammation [9] in humans. We have shown that *F. prausnitzii* administration reduces adipose tissue inflammation [6], but whether XOS be increasing its natural abundance reduces inflammation, is not known. [10,11,12]. In inflammatory conditions, macrophages are typically found around necrotic adipocytes in formations called ‘crown-like structures’ (CLSs) [13]. The density of CLSs has been shown to correlate with obesity, hepatic inflammation and insulin resistance [14,15]. The fecal metabolome gives insight into the GM function and displays certain alterations in metabolic diseases. Fecal short chain fatty acids (SCFAs), trimethylamine, bile acids, ethanol and indole derivatives have risen as potential markers of NAFLD-related dysbiosis of the GM [16]. Decreased fecal SCFAs have been linked to NAFLD [5,17] and increased insulin resistance [18]. In addition, increased SCFA esters, decreased amounts of certain ketones and higher fecal levels of propionate and isobutyrate have been found in patients with NAFLD [19,20]. In our previous study, nuclear magnetic resonance (^1^H-NMR) analysis of cecal metabolites of rats showed that, compared to the HFD, prebiotic XOS reduced cecal levels of isovalerate and tyrosine [5]. Decreased levels of isovalerate have been associated with NALFD in several studies [21,22,23,24] while, similar to XOS, feeding resistant starch to mice on a HFD seemed to also decrease isovalerate levels [25]. Elevated levels of tyrosine have been widely associated with NAFLD [22,24,26,27,28] and, further, its increase can be a marker of progressed steatosis and increased insulin resistance [24,26,27].

Machine learning (ML) models that use the taxonomic composition of highly complex microbial communities to predict host characteristics and disease have been increasingly popular and have helped to uncover novel information on the interactions of our microbiome and health [29,30,31]. The associations between fecal metabolomes, taxonomic composition, and metabolic health are less well characterized, however. Using classification and regression models, we recently explored the GM as a supplement to conventional risk factors to predict prevalent and incident liver disease occurrence and severity with improved accuracy in a large prospective population cohort [32,33]. Biclustering algorithms are a commonly utilized technique in gene expression studies that provide possibilities to detect associations between taxonomic and metabolomic variation by grouping the rows and columns of a 2-dimensional matrix [34,35]. Applications of biclustering in the field of microbiology are encouraging, if still rather sparse [36,37,38]. Moreover, the existing body of research has concentrated on biclustering microbial samples and spectroscopic or spectrometric peaks to taxonomically group the samples and even predict the taxon in a random sample. This knowledge can be extended to multiomic data sets derived from a given biological matrix, where biclustering can be used to identify co-occurring sets of microbial species and their co-varying metabolic signatures.

In this study, we determined whether the effects of XOS on the GM and hepatic health were associated with reduced adipose tissue inflammation as well as improved lipid metabolism and insulin signaling in liver and adipose tissues. To this end, we looked at the phosphorylation levels of key proteins related to insulin signaling and lipid metabolism in liver and adipose tissues. To further characterize adipose tissue inflammation, we measured the common leukocyte antigen CD45 and the density of CLSs. We also determined the expression levels of hepatic microRNAs (miRNA) previously linked to NAFLD [39,40], and the activity of metabolic enzymes in adipose tissues and *gastrocnemius* muscle. Our central hypothesis was that the GM composition and fecal metabolome would display specific signatures of XOS supplementation and thus partly explain the improvement of NAFLD. To this end, we utilized a biclustering algorithm on the correlation coefficient matrix of bacterial genera and metabolites to identify coherent and interpretable, biologically relevant biclusters and reduce effective data dimensionality. Finally, we identified signatures of XOS supplementation and changes in adipose tissue and liver metabolism. Overall, our findings demonstrate how the currently available data integration techniques, such as biclustering, can facilitate joint analysis of multiple parallel omic data types and thus provide novel insights into the interplay between different levels of taxonomic and functional variation in host-associated microbial communities.

## 2. Materials and Methods

### 2.1. Animals, Diets and Analysis of Hepatic Fat Content

An approval for the animal experiment was received from the National Animal Experiment Board of Southern Finland (ESAVI/8805/4.10.07/2017), and the study was performed in accordance with the Guidelines of the European Community Council directives 2010/63/EU and the European Convention for Protection of Vertebrate Animals used for Experimental and other Scientific Purposes (Council of Europe No123, Strasbourg 1985). Four different diets were administered *ad libitum* to rats as described previously [5]. XOS was supplemented or not with high (HFD, 60% of energy from fat, Labdiet/Testdiet, London, UK) or “low” (LFD, 10% of energy from fat, Labdiet/Testdiet) fat diet for 12 weeks in male Wistar rats (*n* = 10/group). The average dose of XOS for the rats in our study was 0.05 g/kg. XOS (95% pure, CAS #87099-0) isolated from corncobs (*Zea mays* subsp. mays) by enzymatic hydrolysis was donated by Shandong Longlive Biotechnology (Yucheng, China). The body weight was determined with an electronic scale before necropsy. At necropsy, liver was excised and weighted with electronic scale. The average body and liver weights as well as the ratio of liver weight/body weight for each diet group are provided in the supplements (Appendix A). The hepatic fat content was determined biochemically and histologically, and the methods and results were presented in our previous publication [5].

### 2.2. Collection and Analysis of Fecal Samples

The contents of the cecum were collected at the time of necropsy, snap-frozen in liquid nitrogen and stored at −80 °C, as described previously [5]. Extraction of the bacterial DNA from the cecum contents, 16S rRNA gene sequencing and processing of the sequence data and extraction of operational taxonomic units (OTUs) were done as described previously [5]. Sample preparation for metabolomic analysis, NMR measurement and identification and analyses of cecal metabolites were done as described previously [5].

Altogether 226 bacterial genera and 38 metabolites were included in the downstream data analyses for this study. Both datasets were explored for features with low prevalence. For the bacterial genera, features with a prevalence of <10% at the 0.1% relative abundance were considered to have low prevalence, which left us with 66 genera. Data loss, calculated as the sum of relative abundance per sample, was on average 0.7%. For the metabolites, features with a prevalence of <10% at 0 absolute abundance were considered to have low prevalence. No metabolites were filtered out.

It should be noted that to minimize redundant fields in the correlation matrix, we filtered out dataset for variables with high amounts of zero or missing values, which was the case for most genera. In the preprocessing phase, we filtered out roughly 70% of the genera, using 10% prevalence at 0.1% relative abundance as the cut off. In this process, the relative abundances of the phyla Firmicutes and Bacteroidetes increased 32 pp and decreased 45 pp, respectively. In previous studies Firmicutes accounted for most taxa predictive of liver disease [32,33] and in our case the dropped genera had low overall significance.

### 2.3. Biclustering

To remove effects bacterial compositionality, we applied the centered-log ratio (clr) transformation to the filtered genus-level abundances. Before the transformation, +1 was added to absolute abundances to avoid division by zero errors. The maximum effect of the +1 addition was no more than 0.07 pp on relative abundances. Spearman correlations were calculated between the clr-transformed genus and raw metabolite abundances. The co-variation in the genus and metabolite abundances was then investigated by forming biclusters on the Spearman correlation coefficient matrix. We used the spectral co-clustering algorithm described by Dhillon [41], available through the spectral coclustering method from the Scikit learn package. The optimal number of clusters was determined to be five by measuring Silhouette and Calinski-Harabasz scores for each model with cluster amounts 2–38, for metabolite and genus axes separately. By functionality, the spectral co-clustering method attempts to form a block-diagonal bicluster structure from the highest values, in this case the strongest positive correlations, with each row and each column belonging to exactly one bicluster. This allows the discovery of meaningful negative correlative patterns by examining cross-cluster correlations.

### 2.4. Preparation of Tissue Protein Homogenates and Measurement of Metabolic Enzyme Activities

The epididymal, mesenteric and subcutaneous adipose tissues, liver and *gastrocnemius* muscle were harvested upon necropsy, snap-frozen in liquid nitrogen and stored until use at −80 °C. After pulverizing the tissues in liquid nitrogen, the tissues were further homogenized in ice-cold lysis buffer using TissueLyser (Qiagen, Valencia, CA, USA). For liver and muscle, the total proteins were extracted from ~25 mg of pulverized tissue using 10 times of volume (*v*/*w*) of buffer that contained 50 mM Tris-HCl (pH 7.4), 150 mM NaCl_2_, 1% NP-40, 1 mM NaVO_4_, 0.1% SDS and 1 mM DTT, supplemented with protease and phosphatase inhibitors (Thermo Fischer Scientific, Waltham, MA, USA). For epididymal, mesenteric and subcutaneous adipose tissues, the total proteins were extracted from ~100 mg of pulverized tissue using 4 times of volume (*v*/*w*) of buffer that contained 10 mM Tris-HCl (pH 7.4), 150 mM NaCl_2_, 2 mM EDTA, 1% Triton-X-100, 10% glycerol and 1 mM DTT, supplemented with protease and phosphatase inhibitors (Thermo Fischer Scientific).

The enzyme activities of citrate synthase (CS), aspartate aminotransferase (AST) and alanine aminotransferase (ALT) in the adipose tissues and *gastrocnemius* muscle protein homogenates were measured with Konelab 20xTi analyzer (Thermo Fischer Scientific) using commercial kits, and the 3-hydroxyacyl-CoA dehydrogenase 8 (β-HAD) activity in a solution that contained 50 mM triethanolamine-HCl (pH 7.0), 4 mM EDTA, 0.04 mM NADH, and 0.015 mM S-acetoacyl-CoA.

### 2.5. Western Blot Analyses of the Phosphorylated Proteins

A total of 60 µg of protein homogenates from liver and 50 µg of total protein from adipose tissues were run on Criterion™ TGX Stain-Free 4–20% gradient gels (Bio-Rad Laboratories, Hercules, CA, USA). After that, the gels were ultraviolet (UV) -activated with ChemiDoc™ imaging system (Bio-Rad) using default settings for the stain free gel activation. Then, the proteins were blotted onto nitrocellulose membranes using Trans-Blot^®^ Turbo™ RTA Midi Nitrocellulose Transfer Kit (Bio-Rad) and Trans-Blot^®^ Turbo™ Transfer System (Bio-Rad). After blotting, the membranes were imaged with ChemiDoc™ imaging system (Bio-Rad) using default settings for the stain free blots. Then, the membranes were cut horizontally in order to separate the molecular weights corresponding to the proteins of interest, and afterwards blocked with Odyssey^®^ Blocking buffer (LI-COR Biosciences, Lincoln, NE, USA) for 1 h at RT. Then, the membranes were incubated overnight at +4 °C with the primary antibodies diluted at 1:1000 in Odyssey^®^ Blocking buffer. All primary antibodies were purchased from Cell Signaling Technology (Danvers, MA, USA). On the next day, the membranes were incubated with the secondary antibody donkey anti-rabbit IRDye 800CW (LI-COR Biosciences) diluted at 1:20,000 in Odyssey^®^ Blocking buffer for 1 h at RT. Finally, the images were acquired with ChemiDoc™ imaging system using default settings for IR Dye 800CW blot. To quantify the phosphorylation levels of the proteins Image Lab 6.0 –software (BioRad) was used. The intensities of the protein bands of interest were normalized to intensities of the stain free blot.

### 2.6. Quantitative Real-Time PCR Analyses

To analyze the expression levels of hepatic miRNAs and *Actb* mRNA, as well as adipose tissue *Cd45* mRNA, the total RNA was extracted from ~100 mg of pulverized epididymal and mesenteric adipose tissues and ~20 mg of pulverized liver by homogenizing with TissueLyser (Qiagen, Germantown, MD, USA) in Trizol reagent (Invitrogen, Carlsbad, CA, USA) according to the supplied protocol. For the analysis of the *Cd45* and *Actb* mRNA, total RNA was reverse transcribed using the High-Capacity cDNA Synthesis Kit (Applied Biosystems, Foster City, CA, USA) according to the instructions of the manufacturer. For the analyses of miRNAs, the total RNA was reverse transcribed using miScript II RT Kit with HiFlex buffer (Qiagen).

Real-time quantitative PCR (qPCR) analysis of *Cd45* and *Actb* mRNA was performed using iQ SYBR Supermix and the CFX96™ Real-Time PCR Detection System (Bio-Rad). The sequences of the in-house designed primers were as follows: *Cd45* forward 5′-CCGTTGTACACCAGAGATGA-3′, *Cd45* reverse 5′-TCCCAAAATCAGTCTGCAC-3′, *Actb* forward 5′-GGCACCACACTTTCTACAAT-3′ and *Actb* reverse 5′-AGGTCTCAAACATGA TCTGG-3′. The expression levels of *Cd45* mRNA were normalized to the quantity of cDNA in the samples that were determined with Quant-iT PicoGreen dsDNA Assay Kit (Invitrogen) according to the manufacturer’s instructions. The fluorescence was detected with GloMax Multi+ microplate reader (Promega Biosystems, Sunnyvale, CA, USA).

To quantify the expression levels of hepatic miRNAs, iQ SYBR Supermix and the CFX96™ Real-Time PCR Detection System (Bio-Rad) were also used. The sequence of the universal primer was 5′-GAATCGAGCACCAGTTACGC-3′. The sequences of miRNA specific primers were obtained from miRBase [42], and were as follows:rno-miR-21-5p MIMAT0000790: 5′-UAGCUUAUCAGACUGAUGUUGA-3′.rno-miR-122-5p MIMAT0000827: 5′-UGGAGUGUGACAAUGGUGUUUG-3′.rno-miR-192-5p MIMAT0000867: 5′-CUGACCUAUGAAUUGACAGCC-3′.rno-miR-221-3p MIMAT0000890: 5′-AGCUACAUUGUCUGCUGGGUUUC-3′.

The expression levels of miRNAs were normalized to the levels of *Actb* because the common miRNA endogenous controls U6 and RNU6B have been shown to be highly variably expressed in liver samples [43].

### 2.7. Histopathological Scoring of the Epididymal Adipose Tissue

The epididymal adipose tissue samples for the histological analysis were fixed in buffered 4% paraformaldehyde for 48 h at +4 °C, transferred into PBS and stored at +4 °C until histological processing. The samples were routinely embedded into paraffin, sectioned at 6 µm and stained with hematoxylin and eosin (H&E). After general histopathological assessment, the total number of CLS per one tissue section (section areas 25–35 mm^2^) in each animal was counted. The histopathological assessment was done blinded to the treatments.

### 2.8. Statistical Analyses

The statistical analyses, except for the GM and their metabolites were performed with IBM SPSS Statistics v26 for Windows (SPSS, Chicago, IL, USA). The main effects of the diet and XOS as well as their interactive effects on the variables were determined using UNIANOVA. If an effect of XOS was found, the statistical significance of the differences between the XOS-supplemented group and HFD or LFD were further analyzed with Mann Whitney U test. The statistical significance was determined at *p* < 0.05. The statistical analyses for bacterial genera and the metabolites were performed with Python, using Scipy and Sklearn packages. Group differences were analyzed with Mann Whitney U test, as with the biomarkers.

We studied the biclusters, diet types, and measured biomarkers further with supervised and unsupervised ML. First, we fitted a classification model with metabolites and genera from the whole dataset and then from each bicluster separately to predict overall diet, diet fat content or XOS ingestion. We used XGBoost [44] as the classifier, as we previously demonstrated its suitability for microbiological dataset [32,33]. Our model was fitted with raw metabolite abundances without standardization, as XGBoost is a decision-tree based classifier. However, genus abundances were clr-transformed [29,45]. The model performance was evaluated by average accuracy and F1 scores from 5-fold cross validation. Principal component analysis, as implemented in the Scikit Learn package, was used to analyze biomarkers with biclusters and individual features.

## 3. Results

### 3.1. Biclustering

Biclustering identified five sets of co-occurring metabolites and bacterial genera (Table 1). For clarity, we named the biclusters based on their metabolite characteristics. Overall, the spectral co-clustering method provided well-defined biclusters visually and in terms of Silhouette and Calinski-Harabasz scores. We found a decent coherence within the biclusters considering the ontology and structure of the metabolites. SCFAs, carbohydrate metabolism markers and nicotinate appeared together in the same bicluster, named hereafter SCFA bicluster, and showed distinct co-variation. These metabolites were consistently more abundant in the feces of LFD rats, as we have described previously [5]. The product bicluster contained valerate, aspartate and isobutyrate, which are products of amino acid fermentation [46], specifically of the branched chain amino acids leucine, isoleucine and valine [47]. Choline and its degradation products, trimethylamine (TMA) and ethanol appeared in the same bicluster, named hereafter TMA bicluster. Amino acids (AA), including BCAAs and aromatic amino acids, were mostly grouped in the AA bicluster. The isovalerate bicluster contained a heterogeneous group of compounds. Tyrosine and isovalerate, the metabolites previously found to decrease with XOS ingestion on the HFD [5], were found in the AA and isovalerate biclusters, respectively. Tyrosine, however, appeared to co-vary with hypoxanthine and methylamine, which were both lower on the XOS-supplemented diets independent of the dietary fat.

Despite some redundancy, we identified functional coherence within the biclusters on the bacterial genus axis as well. The SCFA bicluster contained a significant proportion of gram-positive, known SCFA producers [48,49] and genera linked to lean phenotypes [50,51,52]. These genera were more abundant in the LFD groups. The isovalerate bicluster, on the other hand, contained genera known to thrive in carbohydrate-deficient conditions [53,54,55] but also genera, which encompass apparent opportunistic pathogens [56,57]. These genera were chiefly more abundant in the HFD groups. The AA bicluster seemed to contain at least two gram-negative genera, which have been previously implicated in adipose tissue inflammation and metabolic dysfunction in mice [20,58]. In addition, the total abundance of these genera was decreased by the XOS on the HFD.

We illustrate the biclustering in a heat map, where the rows and columns are arranged according to the number of the bicluster, and the nodes are colored by the magnitude of the Spearman correlation coefficient (Figure 1a). The strongest positive correlations were observed between the metabolites and the genera in the SCFA bicluster, that is, between SCFAs and SCFA-producers. The strongest negative correlations found were between the metabolites in the SCFA bicluster and the genera in the isovalerate bicluster, which contained most genera associated with the HFD. *Escherichia*, *Shigella* and *Parasutterella* genera are known to produce TMA and ethanol from choline [46,59,60]. These genera and metabolites clustered together, suggesting that the biclusters indeed reflect biological significance.

Compared to the HFD, the HFD + XOS group had higher levels of glycine and TMA bicluster genera, particularly *Marvinbryantia* and *Escherichia-Shigella*. The HFD + XOS group also had lower levels of amino acids and slightly lower levels of amino acid degradation products and metabolites in the isovalerate bicluster. The genera in the AA bicluster, particularly *Bilophila* and *Oscillibacter* were decreased with XOS supplementation.

No bicluster-level differences were observed between the LFD and LFD + XOS groups. Feature-wise, XOS supplementation on the LFD was associated with lower hypoxanthine levels and higher methylamine and 1,3-dihydroxyacetone levels. XOS supplementation associated with increased abundances of genera *Bilophila* and GCA-900066575 and decreased abundances of *Alloprevotella*, *Erysipelotrichaceae* UCG-003, *Erysipelatoclostridium*, *Akkermansia* and [Eubacterium] coprostanoligenes group.

We also plotted the hepatic triglycerides along with the total abundances of the genera and metabolites from each bicluster (Figure 1b). The bicluster compositions seemed to clearly associate with hepatic fat content. This was most visible within the SCFA, product and isovalerate biclusters where the separation of the HFD and LFD groups was most prominent. Higher abundance of SCFAs, glycolysis markers and “lean”-type microbes not only associated with a leaner phenotype and “healthier” diet, but also with better hepatic health in terms of fat content.

We used classification to test how well each bicluster reflected the differences between the diet groups. Moreover, we wanted to see whether the variation within some biclusters could be attributable to the dietary fat or XOS supplementation. For comparison, we trained the selected classifier model first with all genera or metabolites in the data set. To study associations of certain diets and biclusters, we then trained the model separately with the features in each bicluster (Appendix A). The raw metabolite abundances and the clr-transformed genus abundances were used as input values.

The prediction accuracy for overall diet groups was low for each run, most likely implying the homogeneity of the LFD and LFD + XOS groups. Using all biclusters, the XOS supplementation was predicted with a modest 50% accuracy, although a slightly higher accuracy was reached by using features from only the AA bicluster. The metabolites in this bicluster were somewhat better predictors of XOS than the genera (62.5% vs. 45% accuracy). These metabolites, ranked by the highest feature importances were hypoxanthine, tyrosine, glutamate, methylamine, isoleucine, formate, alanine, proline, leucine and phenylalanine.

As apparent in the scattergrams in Figure 1b, the dietary fat explained greatly the variance in several features and had a prominent impact on the separation of the groups. The predictive performance for the dietary fat was high, whether using all features or only the features in the SCFA bicluster, with glucose and butyrate comprising 0.975 and 0.025 of the feature importances, respectively. Almost the same accuracy and f1 scores were achieved by using only the features from product and isovalerate biclusters.

### 3.2. The Effects of the Diets on the Hepatic MicroRNA’s

Hepatic miRNAs that have been previously associated with NAFLD [61] were analyzed with quantitative real-time PCR. Based on the univariate analysis of variance, there were no interactive effects of XOS or dietary fat on the levels of miRNAs. However, based on Mann Whitney U test, compared to the HFD, the HFD + XOS had lower hepatic levels of miR-192-5p (*p* = 0.002) and miR-221-3p (*p* < 0.001), and the LFD group had higher levels of miR-21-5p (*p* < 0.001), as well as decreased the levels of miR-192-5p (*p* = 0.001) and miR-221-3p (*p* < 0.001) (Figure 2). On the LFD, XOS supplementation decreased the hepatic levels of miR-192-5p (*p* < 0.001, Figure 2). Compared to the HFD + XOS, the LFD + XOS had higher levels of miR-21-5p and lower levels of miR-192-5p (*p* < 0.001 for both, Figure 2).

### 3.3. The Effects of the Diets on the Phosphorylation of Insulin Signaling and Fatty Acid Oxidation Related Proteins in Liver

The phosphorylation of acetyl-CoA-carboxylase at Ser79 (p-ACC) directs the fatty acid metabolism from lipogenesis to oxidation. Both dietary fat [F (1, 0.368) = 307.4, *p* < 0.001] and XOS supplementation [F (1, 0.006) = 4.6, *p* = 0.039] affected p-ACC (Figure 3). On the HFD, XOS supplementation slightly decreased, and on the LFD increased it. The phosphorylation of ACC was lower on the HFD than on the LFD. In addition, dietary fat and XOS had an interactive effect on p-ACC [F (1, 0.012) = 13.7, *p* = 0.001]. Dietary fat [F (1, 2.8 × 10^−5^) = 6.8, *p* = 0.013], but not XOS, affected the inhibitory Ser636/639 phosphorylation of insulin receptor substrate 1 (p-IRS1) with the phosphorylation levels being slightly lower in the LFD groups (Figure 3). Fat and XOS had an interactive effect on p-IRS1 [F (1, 2.1 × 10^−5^) = 5.7, *p* = 0.022]. Downstream from the IRS1, XOS affected the Thr308 phosphorylation of protein kinase B (p-AKT) [F (1, 4.7 × 10^−6^) = 4.5, *p* = 0.040) increasing it slightly independent of dietary fat (Figure 3). The dietary fat [F (1, 0.006) = 12.0, *p* = 0.001), but not XOS, affected the Thr202/Tyr204 phosphorylation of extracellular signal-regulated kinase (p-ERK) with the phosphorylation levels being lower on the LFD groups (Figure 3).

### 3.4. The Effects of the Diets on the Markers of Adipose Tissue Inflammation

According to the histopathological scoring of the epididymal adipose tissue, both dietary fat [F (1, 140.237) = 29.2, *p* < 0.001) and XOS [F (1, 28.463) = 5.9, *p* = 0.020) decreased the number of CLSs (Figure 4a) but neither affected the number of mononuclear cells (data not shown).

Quantitative PCR of protein tyrosine phosphatase receptor type C, also known as leukocyte common antigen (CD45) mRNA, revealed that the dietary fat but not XOS affected the relative mRNA. The levels were higher in the LFD than HFD groups both in epididymal (CD45-epi) [F (1, 0.011) = 18.1, *p* < 0.001) and mesenteric (CD45-mese) [F (1, 0.071) = 15.1, *p* < 0.001) adipose tissue (Figure 4b).

In both panels, the effects of XOS and fat separately are indicated in the boxes below the graphs. NS denotes non-significant effect. The effects of XOS were further analyzed with Mann Whitney U test, and on the HFD, XOS tended to decrease CD45 mRNA as indicated by the p-value shown in the graph.

### 3.5. The Effects of the Diets on the Phosphorylation of Insulin Signaling, Fatty Acid Oxidation and Lipolysis Related Proteins in the Epididymal and Subcutaneous Adipose Tissue

In the epididymal adipose tissue, the dietary fat decreased [F (1, 0.73) = 24.5, *p* < 0.001] p-ACC (Figure 5a). The dietary fat also decreased [F (1, 0.09) = 6.2, *p* = 0.017] and XOS tended to increase [F (1, 0.051) = 3.5, *p* = 0.070] the lipolysis activating Ser660 phosphorylation of hormone-sensitive lipase (p-HSL) (Figure 5a). XOS subtly increased p-AKT [F (1, 0.054) = 6.9, *p* = 0.012] (Figure 5a). In the subcutaneous adipose tissue LFD had an increasing effect on p-ACC [F (1, 0.151) = 14.2, *p* = 0.001] but XOS had no effect on it (Figure 5b). No effects of the diets on the phosphorylation of AKT or ERK were found in the subcutaneous adipose tissue (Figure 5b).

In both panels *n* = 8–10/group. The graphs show the quantification of the of the proteins by western blot. The black dots in the bars show individual data points. Above the graph examples of the blot images are shown. The effects of XOS and dietary fat (FAT) separately are indicated in the boxes below the graphs. NS denotes non-significant effect. The effects of XOS were further verified with Mann Whitney U test, and * indicates statistically significant difference between the HFD or LFD group and the corresponding XOS-supplemented group.

### 3.6. The Effects of the Diets on the Activities of Metabolic Enzymes

The LFD increased the activity of AST in the epididymal adipose tissue [F (1, 0.002) = 8.7, *p* = 0.006], while XOS decreased it in the mesenteric adipose tissue [F (1, 0.411) = 7.9, *p* = 0.009] (Figure 6a). The dietary fat and XOS had an interactive effect on AST activity in the subcutaneous adipose tissue [F (1, 0.01) = 12.4, *p* = 0.001] (Figure 6a). XOS supplementation increased the activity of ALT in the epididymal adipose tissue [F (1, 0.006) = 4.7, *p* = 0.018] independent of dietary fat (Figure 6b). An interactive effect of fat and XOS was found on the activity of beta-HAD in the mesenteric adipose tissue [F (1, 9.333) = 14.5, *p* = 0.001], while in the subcutaneous adipose tissue only dietary fat had an increasing effect on it [F (1, 0.027) = 4.7, *p* = 0.037] (Figure 6c). LFD increased the activity of CS in the epididymal adipose tissue [F (1, 0.024) = 17.2, *p* < 0.001], while in gastrocnemius muscle the dietary fat and XOS had an interactive effect on its activity [F (1, 6.649) = 4.7, *p* = 0.037] (Figure 6d).

In all panels *n* = 8–10/group. The graphs show the biochemically measured enzymatic activities. The black dots in the bars show individual data points. The effects of XOS and dietary fat (FAT) separately are indicated in the boxes below the graphs. NS denotes non-significant effect. The effects of XOS were further verified with Mann Whitney U test, and * indicates statistically significant (p < 0.05) difference between the HFD or LFD group and the corresponding XOS-supplemented group 

### 3.7. Associations between the Biclusters and Biomarkers

We investigated how the total abundances of the metabolites (Figure 7a) or genera (Figure 7b) in each bicluster were associated with the biomarkers of liver or adipose tissue. Most notably miR-221-3p, miR-192-5p, hepatic triglycerides, CLSs and, to a lesser amount, p-IRS1-liver and p-ERK-liver correlated negatively with SCFAs, carbohydrate metabolism markers (*p* < 0.05 for all, Figure 7a), and the “lean-type” genera in the SCFA bicluster (*p* < 0.05 for all, Figure 7b). These biomarkers were positively associated to metabolites and genera in the product bicluster and genera in the isovalerate bicluster.

Conversely, p-ACC-liver, CS-epididymal, AST-mesenteric, CD45 and miR-21-5p, were positively associated with features in the SCFA bicluster and negatively associated with features in the product bicluster and genera in the isovalerate bicluster (*p* < 0.05 for all, Figure 7a,b). None of the biomarkers were significantly associated with the total abundances of metabolites in the TMA bicluster, however, AST-subcutaneous, miR-21-5p and AST-epididymal were negatively and CLSs positively associated with the TMA bicluster genera (*p* < 0.05 for all, Figure 7b).

The total abundance of the metabolites in the isovalerate bicluster was negatively associated with p-HSL-epididymal, p-AKT- epididymal and p-ACC- epididymal (*p* < 0.05 for all, Figure 7a). In addition, feature-wise observations revealed that isovalerate, which was significantly elevated in the HFD group compared to the HFD + XOS group, was positively associated with hepatic triglycerides, CLSs, miR-221-3p and miR-192-5p (*p* < 0.05 for all). Isovalerate was also negatively associated with miR-21-5p, p-ACC-liver, CD45-epididymal, CS-epididymal, p-ACC-epididymal and p-AKT-epididymal (*p* < 0.05 for all).

To study whether different co-variation of the metabolites, genera and biomarkers were linked to the effects of XOS, we further analyzed the associations within the HFD and HFD + XOS groups by principal component analysis (PCA). For the comparison, we used the total metabolite or total genus abundances of each bicluster and biomarkers with significant or visually interesting differences between the two groups. The groups differed markedly along the first principal component, containing 20.5% of the total variance (Figure 8). Most of this variance was explained by miR-192-5p, miR-221-3p, AA bicluster, isovalerate bicluster, p-AKT-epididymal, p-ERK-liver and p-HSL-liver, as apparent by the loadings (Figure 8).

The total metabolite abundances in the AA and isovalerate biclusters positively co-varied with miR-192-5p, miR-221-3p and to a lesser amount with AST-mesenteric, CLSs and hepatic triglycerides. Conversely, they co-varied negatively with p-ERK-liver and p-HSL-epididymal. The genera in the AA bicluster tended to co-vary with CLSs and hepatic triglycerides.

In the PCA of significantly different features (Figure 9), the group separation was observed along explained 31.4% of the total variance. Methylamine co-varied strongly and positively with hepatic miRNAs and AST-mesenteric and negatively with pATK-epididymal and ALT-epididymal. Hypoxanthine co-varied with CLSs, hepatic triglycerides and p-AKT-epididymal. Noticeably, BCAAs (leucine, isoleucine, and valine) and aromatic amino acids (phenylalanine and tyrosine) were clustered with triglycerides along the first component, which explained most of the between-group variation, whereas glycine, along with ALT-epididymal, distinctly co-varied along the second component. The genera *Oscillibacter*, *Bilophila* and *Ruminiclostridium*
*5*, which were significantly decreased with XOS (Figure 10), were also associated with hepatic triglycerides and miRNAs. *Escherichia-Shigella* and *Marvinbryantia* were increased with XOS supplementation and co-varied positively with p-AKT-epididymal and negatively with CLSs, hepatic triglycerides and miRNAs.

## 4. Discussion

In the present study, we sought to characterize the underlying tissue-specific mechanisms associated with the steatosis-preventing effects of XOS supplementation in rats. In addition, we implemented novel bioinformatic analyses to understand whether the interplay between the gut microbes and their metabolites could explain the prebiotic and hepatic health-promoting effects of XOS. Indeed, the hepatic steatosis-preventing effects of XOS were associated with decreased adipose tissue inflammation, likely improved insulin signaling and certain clusters of co-occurring fecal metabolites and bacterial genera.

The development of biclustering techniques has been driven by the need to capture relevant biclusters, that relate to real biological associations and physical conditions. Much of this development has been in the context of transcriptomics [62], but biclustering techniques can potentially benefit other omics studies as well, such as discovering the biological co-occurrence of microbes and their metabolites [36]. Here we utilized the spectral co-clustering method [41] on a data table of correlations between the metabolites and bacterial genera and demonstrated its utility in (a) Identifying co-occurring bacterial genera and subsets of relevant metabolic conditions and (b) Thus reducing the effective dimensionality in the analysis of a multiomic dataset and improving interpretability.

The identified biclusters captured that the SCFAs and co-varying markers of carbohydrate metabolism were decreased on the HFD, which is in agreement with decreased SCFAs having been previously linked to NAFLD and insulin resistance [5,17,18]. Among the genera that were most abundant on the LFD, in SCFA-rich conditions, were *Anaerostipes* and *Faecalibaculum*, which are known saccharolytic butyrate producers [48,49]. The only known species of *Faecalibaculum*, *F. rodentium,* has also been identified as anti-tumourigenic in mice [48]. The HFD was characterized by lower fecal SCFAs and correlated with an increased abundance of a heterogeneous group of genera. Among these, *Candidatus Soleaferrea* has been described in anorexic subjects [53] and GCA-900066575 and *Lachnoclostridium* in high fat [54] conditions suggesting that our findings might be related to the carbohydrate-deficient conditions on the HFD. We also found choline and its degradation products ethanol and TMA along with known TMA-producing proteobacteria *Escherichia*, *Shigella* and *Parasutterella* [46] in a shared bicluster. Thus, these biclusters seemed to reflect biological significance.

With respect to biclusters, the effects of XOS supplementation seemed to vary depending on the dietary fat, which was apparent by the low classification accuracy. On the LFD, the addition of XOS associated with only a few individual metabolites and genera not visible at the bicluster level. On the HFD, glycine was elevated with XOS supplementation. Impaired glycine biosynthesis has been linked with hyperlipidemia and steatohepatitis and, conversely, administration of glycine in a tripeptide form to mice alleviated both conditions [63]. Notably, the levels of hypoxanthine were decreased by XOS independent of dietary fat. Hypoxanthine is a major substrate of xanthine oxidase, which converts it to uric acid and reactive oxygen species [64]. Xanthine oxidase is highly expressed in liver and recently investigated as a therapeutic target in NAFLD [65]. Reducing oxidative stress through suppression of the xanthine metabolic pathway ameliorated hepatic steatosis and inflammation. Thus, some of the steatosis-reducing effects of XOS might be attributed to higher glycine and lower hypoxanthine levels in our study. Yet, further mechanistic in vitro or in vivo studies will be needed to understand how these metabolites could affect hepatic metabolism.

On the HFD, XOS-supplemented rats had significantly lower levels of isovalerate, methylamine and slightly lower levels of the co-occurring genera. In a recent study, feeding resistant starch to mice on a HFD decreased the levels of isovalerate and abundances of *Oscillibacter, Ruminiclostridium 5* and *Ruminiclostridium 9* [25] being in line with our findings showing that *Oscillibacter* and *Ruminiclostridium 9* were found to co-vary together and with the amino acid levels.

XOS-supplemented rats also had lower levels of BCAAs and aromatic amino acids. This metabolic fingerprint was visible in the biclusters, as these metabolites were contained in AA and isovalerate biclusters. In addition, amino acid fermentation products (the Product bicluster) tended to be lower with XOS. Obesity and NAFLD have been shown to have an interactive effect on serum amino acid levels, increasing BCAAs and aromatic amino acids while decreasing glycine [26,27]. In addition, the same amino acid profile has been linked to metabolic syndrome independent of obesity [66]. We observed a reverse effect by XOS on these amino acids in feces, which raises the question and need for future mechanistic studies to determine whether altered protein catabolism by the GM could be tailgated by insulin resistance and hepatic steatosis.

As described, we aimed to determine the possible relations of the tissue level molecular mechanisms in concert with their interactions with the metabolites and microbes to the hepatic steatosis-preventing effects of XOS. We first analyzed the hepatic expression levels of several miRNAs previously associated with NAFLD [61]. Supporting the role of up-regulated miR-21-5p in reducing lipid accumulation [67], we found that the HFD groups expressed significantly less miR-21-5p, which has been shown to also promote hepatic insulin resistance and steatosis [67,68]. However, contradicting results of serum and hepatic miR-21-5p levels have been reported in both humans and in animal models of NAFLD/NASH. Some studies show an up-regulation of miR-21-5p [69,70], while others report a down-regulation [67,71]. miR-122-5p is involved in the regulation of hepatic lipid metabolism among other several physiological processes in hepatic function [72]. In our study, there were no differences in miR-122-5p levels between the groups suggesting that miR-122-5p is not responsive to HFD or XOS in rat liver. XOS lowered the levels of hepatic miR-192-5p both on the HFD and LFD, which is contradictory to rodent studies showing that NAFLD or HFD decreased hepatic miR-192-5p expression leading to lipid accumulation in cells [40,73]. It is possible that our results with XOS-induced miR-192-5p down-regulation could relate to reduced hepatic inflammation [74], however, further studies are needed to determine this effect of miR-192-5p. In previous studies miR-221-3p has been shown to be involved in NASH-induced carcinoma mouse model [75]. Furthermore, up-regulation of miR-221-3p in liver has been observed in NASH patients in a fibrosis-dependent manner as well as in a mouse model of hepatic fibrosis [76]. Our results show that XOS down-regulated hepatic miR-221-3p on the HFD suggesting a healthier hepatic miR-profile in response to XOS supplementation. However, in our model fibrosis was not present as reported previously [5]. In our study, miRs 192-5p and 221-3p were associated with hepatic p-ACC and triglycerides, supporting the emerging role of these miRs in regulating lipid metabolism [39,40]. Nevertheless, the mechanistic role of these miRs in hepatic fat accumulation remains to be resolved in future studies.

On the HFD, XOS slightly decreased the hepatic levels of Ser79-phosphorylated ACC, the enzyme that catalyzes the rate-limiting step of de novo lipogenesis (Wakil et al. 1983). The phosphorylation of ACC at Ser79 inactivates ACC inhibiting the production of malonyl-CoA, which causes a decrease in de novo lipogenesis [77,78,79,80] and enhances fatty acid β-oxidation [80,81]. Based on our results, the prevention of hepatic fat accumulation by XOS did not involve enhanced ACC activation by Ser79 phosphorylation. However, we found that higher fecal glucose, nicotinate (niacin) and butyrate along with SCFA-producing bacterial genera, particularly *Faecalibaculum*, were highly predictive of higher p-ACC. This is not surprising considering that SCFAs and niacin have both been shown to impede hepatic de novo lipogenesis [82,83].

Upon binding of insulin, IRS1 subsequently recruits phosphoinositide 3-kinase (PI3K), which activates AKT by phosphorylating it at Thr308 [84]. While the tyrosine phosphorylation of IRS1 promotes its activity, serine phosphorylation of IRS1 inhibits its functions [85]. We did not find effects of XOS on the Ser-phosphorylation of IRS1, yet, as expected, the LFD groups expressed less phosphorylated IRS1 indicating better insulin signaling compared to the HFD groups. However, downstream of IRS1, XOS enhanced the phosphorylation of AKT independent of dietary fat in the epididymal adipose tissue and liver. This is in agreement with a recent study in type 2 diabetic rats showing that XOS increased insulin signaling in muscle [86]. An improved insulin signaling by XOS could be linked to its anti-hyperglycemic effects that were recognized already 30 years ago [87]. Nevertheless, in our previous study XOS did not affect serum glucose levels in the same rats [5]. AKT phosphorylation in the epididymal fat negatively correlated with the genera and metabolites in the isovalerate bicluster, likely reflecting the effects of dietary fat. These associations might relate to the role of BCFAs in regulating adipocyte insulin signaling [88].

Because the pathogenesis of NAFLD is associated with adipose tissue dysfunction [8,9], we also studied whether the XOS-induced reduction in hepatic fat content was linked to alterations in the activity of metabolic enzymes and inflammation in the adipose tissues of the rats. Several studies have shown that in obesity, infiltration of immune cells into the adipose tissue increases inflammation, which is associated with disturbed insulin signaling [89,90,91,92]. Insulin resistance in turn is a risk factor for the onset of NAFLD [93]. We hypothesized that XOS-induced increase of anti-inflammatory *F. prausnitzii*, reduced hepatic fat content and increased AKT phosphorylation would associate with decreased HFD-induced inflammation in the adipose tissue. The epididymal adipose tissue samples exhibited no over pathological changes and the amount of CLS was relatively low in all groups. Highly likely, the adipose tissue macrophages have played a role in our study to contribute to the hepatic fat content because both the XOS and LFD were associated with a reduced number of CLSs in the epididymal adipose tissue. Adipose tissue-resident macrophages are known to importantly contribute to the onset of NAFLD by increasing macrophage recruitment and inflammation in the liver [9,11,12]. In our data, SCFAs and co-varying bacterial genera were associated with lower number of CLSs. SCFAs butyrate, propionate and acetate are known to reduce inflammation in both liver and adipose tissue [16], and thus, their co-variance in our study could reflect biological significance.

To analyze inflammatory factors further, we also determined the mRNA expression levels of CD45 mRNA in the mesenteric and epididymal adipose tissue. CD45 is a transmembrane protein that is present on all leukocytes and their hematopoietic progenitors [94]. In both the epididymal and mesenteric adipose tissue, the expression of CD45 was highest in the LFD groups despite the decreased number of macrophages. This finding was surprising because we have shown that increasing *F. prausnitzii* abundance in mice by oral administration was followed by a decrease in hepatic fat content and a concomitant decrease of CD45-positive leukocytes in the adipose tissue [6]. However, it should be noted that while CLS are macrophages and mainly bone marrow-derived CD45-positive cells [95], based on their cell morphology they are highly differentiated and phagocytize adipocytes [96]. Thus, it might be that the production of CD45 mRNA in these cells is low despite the high protein content. In addition, interestingly, on a contrary to many studies linking immune cell infiltration into the adipose tissue inflammation and NAFLD, some studies have shown that enrichment of specific immune cells in the adipose tissue can in fact prevent the onset of insulin resistance. For instance, Harmon et al. showed that B-1b lymphocytes that are enriched in the visceral adipose tissue of obese humans, decrease inflammation and insulin resistance in the visceral adipose tissue in diet-induced obese mice [97]. In another study, perforin-positive dendritic cells reduced inflammation in the adipose tissue by reducing the number of tissue resident, inflammatory T cells in mice [98]. The results of these two studies raise the question whether some type of immune cells in the adipose tissue may sometimes be a protective instead of detrimental.

Altogether, in the present study, we identified several inflammatory and metabolic changes as well as co-variances of metabolites and bacterial genera underlying the XOS-prevented hepatic steatosis in rats. This validated the use of biclustering as a useful algorithmic tool to assess the biological importance of microbe-metabolite co-occurrence in health and disease.

## 5. Conclusions

Based on our observations, the joint analysis of taxonomic and metabolomic patterns can be supported by biclustering algorithms, which provide a useful technique for detecting interpretable groupings of co-varying microbes and metabolites, potentially linked with health and disease. The preventive effects of XOS on hepatic steatosis could be linked to reduced adipose tissue inflammation, as apparent by the lower counts of CLSs in the epididymal adipose tissue. This reduction was accompanied by increased phosphorylation of AKT in both liver and epididymal adipose tissue, suggesting that less inflammation was associated with improved insulin signaling. On the HFD, XOS increased fecal glycine and decreased BCAAs, aromatic amino acids, hypoxanthine and isovalerate that due to their known functions could partly explain steatosis-reducing effects of XOS. However, future mechanistic studies are needed to understand how these GM-produced metabolites could exactly affect hepatic metabolism and fat content.

## Figures and Tables

**Figure 1 ijerph-18-04049-f001:**
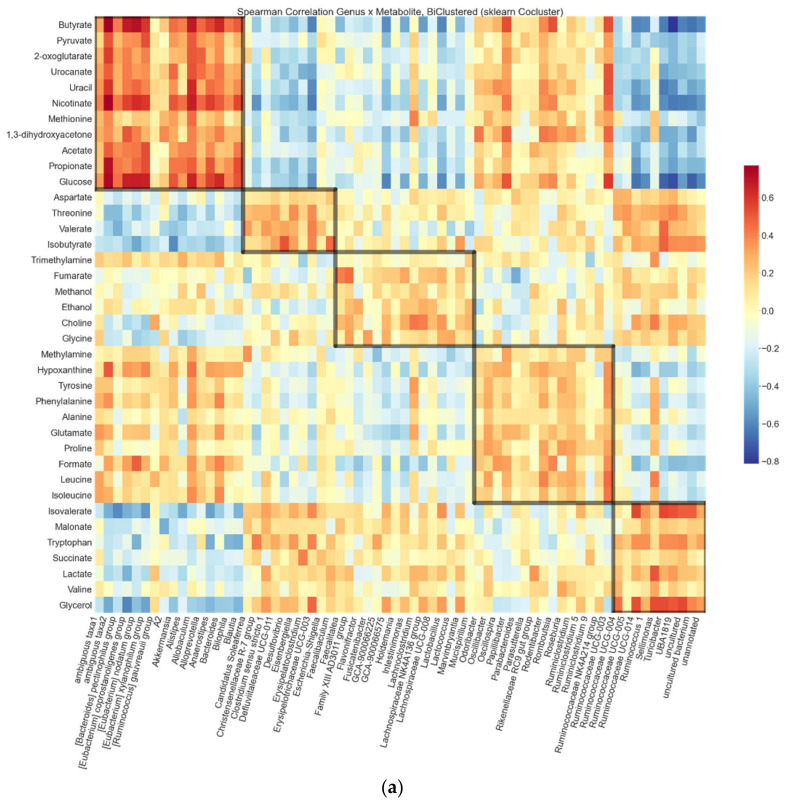
Biclustering results: (**a**) Heat map of Spearman correlation coefficients between the metabolites and genera. The color of a node indicates the magnitude and the direction of correlation. Biclusters suggested by the spectral co-clustering algorithm are visible as the diagonal highlighted blocks. The resulting checkerboard structure is a side product of the algorithm and allows the observation of cross-cluster associations such as the congregated negative correlations in the upper-right corner. (**b**) Summed relative genus and absolute metabolite abundances in each bicluster. Boxes indicate quartiles; points outside whiskers are outliers. Scatter markers were sized to indicate hepatic fat (triglycerides) in each sample. Bacterial relative abundances were summed and then centered log ratio (clr) -transformed. Both axes were mean centered at zero.

**Figure 2 ijerph-18-04049-f002:**
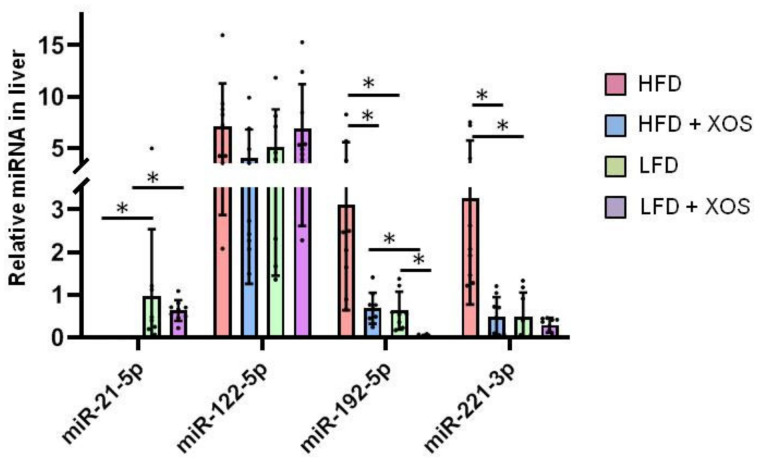
The relative expression levels of miRNAs in liver. *n* = 8–10/group. The graph shows the quantification of the miRNAs by real-time quantitative PCR. The black dots in the bars show individual data points. * indicates statistically significant (*p* < 0.05) difference between the groups as determined by Mann Whitney U test.

**Figure 3 ijerph-18-04049-f003:**
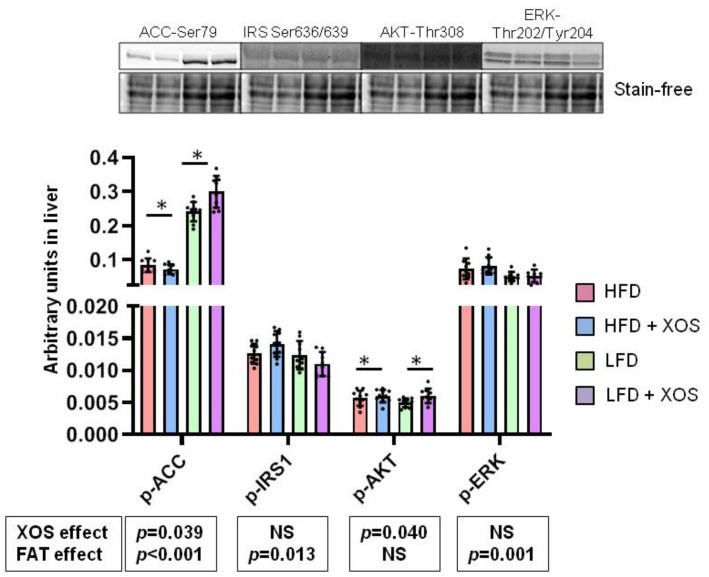
The phosphorylation levels of ACC, IRS1, AKT and ERK in liver. *n* = 8–10/group. The graph shows the quantification of the proteins by Western blot. The black dots in the bars show individual data points. Above the graph examples of the blot images are shown. The effects of XOS and dietary fat (FAT) on each phosphorylated protein are indicated in the boxes below the graphs. NS denotes non-significant effect. The effects of XOS were further verified with Mann Whitney U test, and * indicates statistically significant (*p* < 0.05) difference between the HFD or LFD group and the corresponding XOS-supplemented group as determined by Mann Whitney U test.

**Figure 4 ijerph-18-04049-f004:**
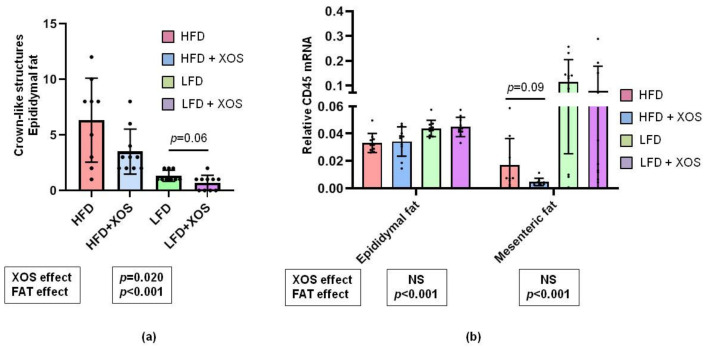
The effects of the diets on adipose tissue inflammation markers. (**a**) The graph shows the observed counts of CLSs in epididymal adipose tissue as determined by histopathological scoring. *N* = 10/group. The black dots in the bars show individual data points. (**b**) The graph shows the relative expression levels of CD45 mRNA in both epididymal and mesenteric adipose tissue that were quantified with qPCR. *n* = 8–10/group. The black dots in the bars show individual data points.

**Figure 5 ijerph-18-04049-f005:**
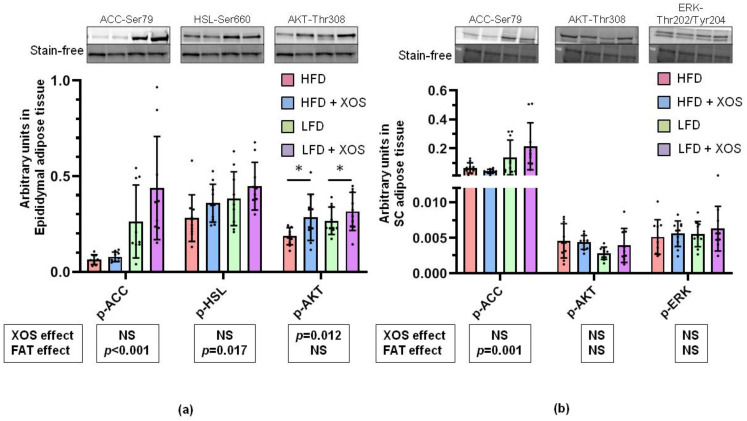
The effects of the diets on the phosphorylation levels of proteins in the adipose tissues. (**a**) Phosphorylation levels of ACC, HSL and AKT in epididymal adipose tissue. (**b**) Phosphorylation levels of ACC, AKT and ERK in subcutaneous (SC) adipose tissue. * indicates statistically significant (*p* < 0.05) difference between the HFD or LFD group and the corresponding XOS-supplemented group as determined by Mann Whitney U test.

**Figure 6 ijerph-18-04049-f006:**
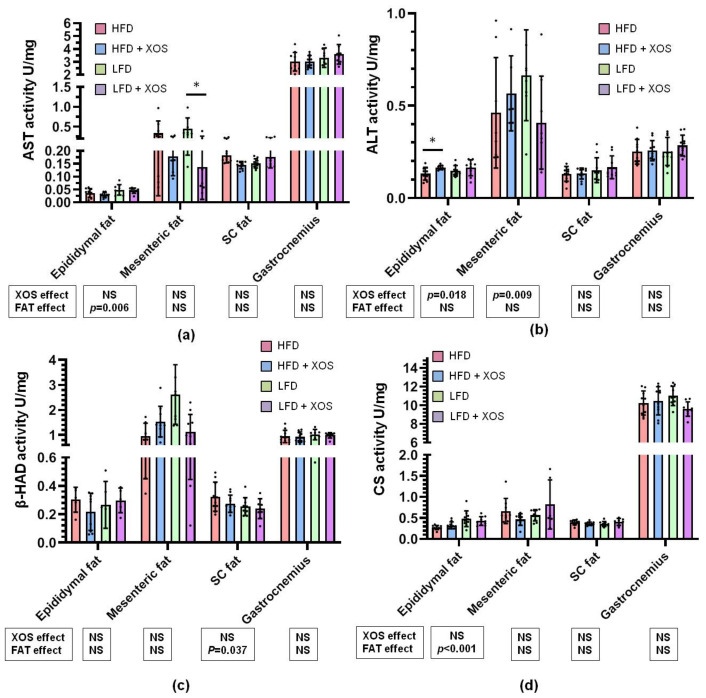
The activities of (**a**) AST, (**b**) ALT, (**c**) β-HAD and (**d**) CS in epididymal, mesenteric and subcutaneous (SC) adipose tissue as well as *gastrocnemius* muscle. * indicates statistically significant (*p* < 0.05) difference between the HFD or LFD group and the corresponding XOS-supplemented group as determined by Mann Whitney U test.

**Figure 7 ijerph-18-04049-f007:**
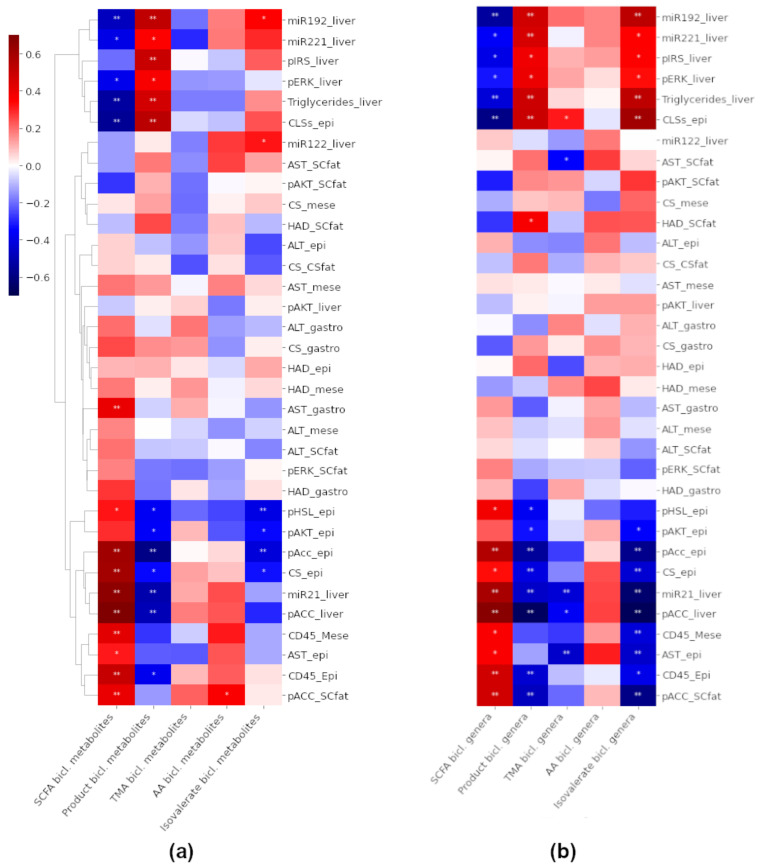
Heat maps of Spearman correlation coefficients between the total abundances of (**a**) The metabolites and (**b**) The genera in each bicluster and different biomarkers. The color of a node indicates the magnitude and the direction of correlation. Hierarchical clustering was applied to biomarkers. The raw metabolite abundances were summed, and the relative genus abundances were summed and then clr-transformed. epi = epididymal fat, mese = mesenteric fat, SCfat = subcutaneous fat, gastro = *gastrocnemius* muscle. * denotes *p* < 0.05 and ** *p* < 0.01.

**Figure 8 ijerph-18-04049-f008:**
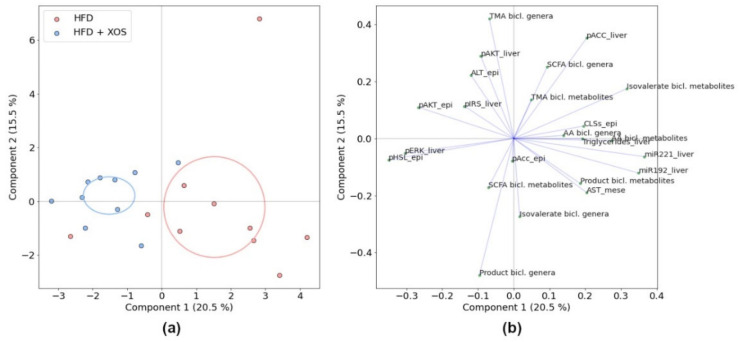
PCA of the HFD and HFD + XOS groups using total feature abundances in each bicluster and selected biomarkers, (**a**) Ordination and (**b**) Loading plots. The total explained variance for each component is shown in parentheses, and the ellipses indicate 95% confidence intervals.

**Figure 9 ijerph-18-04049-f009:**
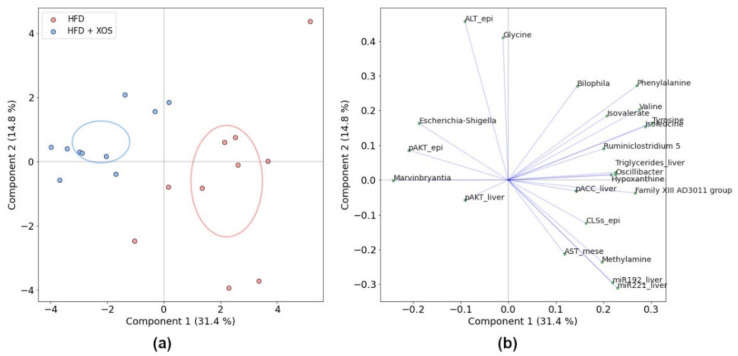
PCA of the HFD and HFD + XOS groups with all significantly differing features and biomarkers, (**a**) Ordination and (**b**) Loading plots. The ellipses indicate 95% confidence intervals. The total explained variance for each component is shown in parentheses. epi = epididymal fat, mese = mesenteric fat.

**Figure 10 ijerph-18-04049-f010:**
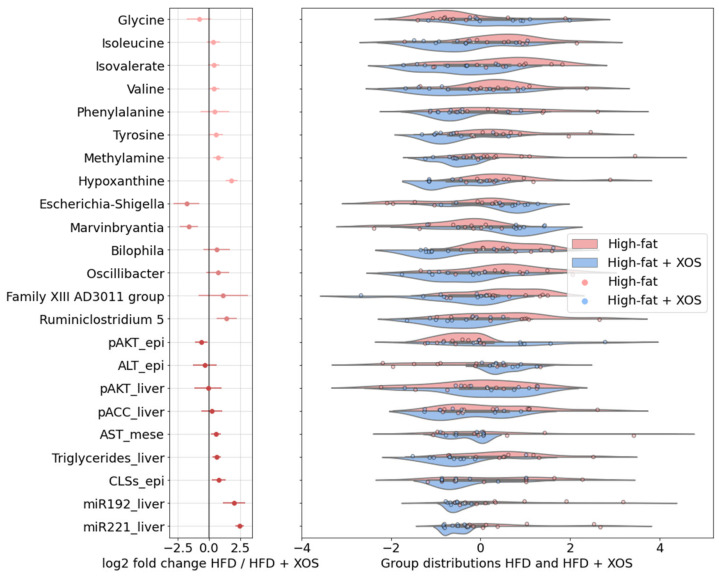
The features with significant differences between the HFD and HFD + XOS groups, log_2_ fold changes and mean-centered values showing group distributions. Negative log_2_ fold change indicates higher log_2_ effect size with XOS, and positive indicates higher log_2_ effect size without XOS. Bars indicate error. Kernel density bandwidth of the violin plot 0.5. epi = epididymal fat, mese = mesenteric fat.

**Table 1 ijerph-18-04049-t001:** The features contained in each bicluster, with non-annotated bacterial genera excluded. Positive log_2_ fold change indicates higher abundance in the HFD group.

Bicluster	Genera in Bicluster	Log2 Fold Change HFD/HFD + XOS	Log2 Fold Change HFD/LFD	Metabolites in Bicluster	Log2 Fold Change HFD/HFD + XOS	Log2 Fold Change HFD/LFD
SCFA Bicluster	[Ruminococcus] gauvreauii group	–0.14	–2.43 *	1,3-dihydroxyacetone	0.17	–0.61
	Anaerostipes	–1.41	–4.04 **	2-oxoglutarate	0.27	–0.44
	Christensenellaceae R-7 group	0.42	–0.2	Acetate	–0.05	–0.52 **
	Clostridium sensu stricto 1	–0.23	–3.78 *	Butyrate	–0.07	–1.51 **
	Faecalibaculum	1.48	–5.69 **	Glucose	0.11	–1.96 **
	GCA-900066225	–0.81	–2.47 **	Methionine	0.38	–0.06
	Papillibacter	0.86	–3.87 **	Nicotinate	0.05	–1.06 **
	Parabacteroides	0.29	–0.48 *	Propionate	–0.01	–0.62 **
	Rodentibacter	–0.83	–2	Pyruvate	0.63	–0.98 *
	Romboutsia	1.69	–0.75	Uracil	0.59	–0.51 *
	Ruminiclostridium 5	1.39 *	–2.31 **	Urocanate	0.48	–0.63
	Ruminococcaceae UCG-014	0.25	–0.82			
	Turicibacter	1.16	–3.96 **			
Product Bicluster	[Bacteroides] pectinophilus group	–3.51	0.8	Aspartate	0.06	0.17
	[Eubacterium] coprostanoligenes group	–0.42	–0.12	Isobutyrate	0.08	0.57
	[Eubacterium] xylanophilum group	0.81	0.66	Threonine	0.09	0.47 *
	Akkermansia	–0.29	0.21	Valerate	0.25	0.47 *
	Alistipes	–0.01	1.13 *			
	Defluviitaleaceae UCG-011	0.37	2.38 **			
	Family XIII AD3011 group	1.13 **	1.33 *			
	Odoribacter	–0.33	1.35			
	Ruminococcaceae UCG-005	0.77	1.93			
TMA Bicluster	Allobaculum	0.93	1.6	Choline	0.97	1.45 *
	Alloprevotella	–0.5	–0.05	Ethanol	–0.2	–0.08
	Bacteroides	–0.1	0.38	Fumarate	0.01	–0.39
	Blautia	–0.25	–0.69	Glycine	–0.81 *	0.48
	Erysipelatoclostridium	–1.31	–0.04	Methanol	–1.09	0.24
	Escherichia-Shigella	–1.82 **	–0.93	Trimethylamine	0.15	0.11
	Faecalitalea	–0.65	1.24 *			
	Flavonifractor	0.96	3.07 *			
	Fusicatenibacter	3.88	10.07			
	Holdemania	–0.32	0.9			
	Intestinimonas	–0.18	–0.57			
	Lachnospiraceae UCG-008	–0.82	1.46			
	Marvinbryantia	–1.65 **	–0.95			
	Oscillospira	–0.83	8.89			
	Parasutterella	0.63	0.98			
AA Bicluster	Bilophila	0.61 *	3.41 **	Alanine	0.19	0.04
	Desulfovibrio	0.75	–0.25	Formate	–0.8	–0.75 **
	Eisenbergiella	–0.03	0.05	Glutamate	0.08	–0.03
	Erysipelotrichaceae UCG-003	–0.23	–1.25	Hypoxanthine	1.8 **	–0.46 *
	Lachnospiraceae NK4A136 group	1.27	0.87	Isoleucine	0.34 *	–0.16
	Mucispirillum	1.51	1.03	Leucine	0.31	0.07
	Oscillibacter	0.7 *	–0.97 *	Methylamine	0.73 **	0.5
	Roseburia	–0.08	–1.42	Phenylalanine	0.45 *	–0.04
	Ruminiclostridium	1.27	–0.06	Proline	0.33	0
	Ruminiclostridium 9	0.55	0.31	Tyrosine	0.55 **	0.02
	Ruminococcaceae UCG-003	0.67	0.18			
	Ruminococcaceae UCG-004	–2.8	–2.33			
	Ruminococcus 1	0.13	0.35			
Isovalerate Bicluster	[Eubacterium] nodatum group	0.36	3.31 **	Glycerol	1.57	3.0 **
	Candidatus Soleaferrea	0.58	2.21 **	Isovalerate	0.37 *	1.02 **
	GCA-900066575	0.41	1.0 *	Lactate	0.63	0.9
	Lachnoclostridium	–0.33	2.38 *	Malonate	–0.11	0.28
	Lactobacillus	0.28	0.51	Succinate	–0.35	–1.53
	Lactococcus	0.54	0.6	Tryptophan	–0.11	0.45
	Rikenellaceae RC9 gut group	0.31	1.43 *	Valine	0.39 *	0.09
	Ruminococcaceae NK4A214 group	–0.04	1.91 **			
	Sellimonas	0.57	2.57 **			
	UBA1819	–0.63	1.86 **			

Mann-Whitney U: * *p* < 0.05, ** *p* < 0.01.

## Data Availability

The data presented in this study are openly available in figshare at https://doi.org/10.6084/m9.figshare.14112908.v1 (accessed on 8 April 2021).

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
