# Peer review of "Xylo-Oligosaccharides in Prevention of Hepatic Steatosis and Adipose Tissue Inflammation: Associating Taxonomic and Metabolomic Patterns in Fecal Microbiomes with Biclustering"

_ijerph, 2021, doi:10.3390/ijerph18084049_

Round 1

Reviewer 1 Report

The authors describe lab experiment detailing the role of Xylo-oligosaccharides (XOS) in preventing hepatic steatosis. The results are very interesting and of great scientific importance. Minor changes are recommended here to help improve the manuscript.

1. The introduction is very long and diverges from the main issue. Please reduce the length of the introduction with the focus on XOS and the use of biclustering as an algorithmic method. Paras 2, 3, and 4 can be combined into one with a skillful reduction in volume. 

Author Response

Review points and responses

  1. The introduction is very long and diverges from the main issue. Please reduce the length of the introduction with the focus on XOS and the use of biclustering as an algorithmic method. Paras 2, 3, and 4 can be combined into one with a skillful reduction in volume. 

We thank the reviewer for this valuable suggestion. We have compressed the intro into 4 relevant paragraphs (lines 32 – 109) and combined the suggested parts. We believe it now provides a more concise introduction into the concept without being too exhausting to read.

Reviewer 2 Report

Manuscript title: Xylo-oligosaccharides in prevention of hepatic steatosis and adipose tissue inflammation: associating taxonomic and metabolomic patterns in fecal microbiomes with biclustering

In the manuscript, the authors investigated the effects of xylo-oligosaccharides (XOS) on the metabolomic patterns in fecal microbiomes in HFD-induced hepatic steatosis with biclustering algorithmic method. In general, the authors have completed a reasonable study with very rich and informative algorithmic data of metabolomics in HFD-induced symptoms and indicated the biomarkers in the ameliorative results. In particular, the authors have carefully conducted research in the statistical analysis of the differences between the trial subjects. However, a few of minor concerns would be suggested and requested for further improvement in the manuscript.

  1. Materials and Methods
    The source and composition of XOS material should be presented in detail in the section, because of a lot of product variety in the market.

  1. Results
    The histological analysis for proving the hepatic steatosis and the preventive effects of fatty liver in the animal study should be presented for further demonstration.

  1. Please rephrase the sentence:
    Line 466-The dietary fat and XOS had an interactive effect on the activity on AST activity in the subcutaneous adipose tissue.

  1. Line 540. Oscillibacter, Bilophila and Ruminiclostridium 5, which were significantly decreased with XOS
    Please check the Ruminiclostridium 5.

Author Response

Review points and responses

1. Materials and Methods
The source and composition of XOS material should be presented in detail in the section, because of a lot of product variety in the market.

Details and origin of the XOS used in our study are now described on lines 120-123: “The average dose of XOS for the rats in our study was 0.05 g/kg. XOS was isolated from corncobs (Zea mays subsp. mays) by hydrolyzing enzymatically and donated by Shan-dong Longlive Biotechnology (95% pure, CAS #87099-0).”

2. Results
The histological analysis for proving the hepatic steatosis and the preventive effects of fatty liver in the animal study should be presented for further demonstration.

The hepatic fat content results were already published before. We apologize for not having that described clearly in the previous version of the manuscript. We now direct the reader to our previous open access article where we have provided the methods and results of the analysis of liver triglycerides (lines 126-128)

3. Please rephrase the sentence:
Line 466-The dietary fat and XOS had an interactive effect on the activity on AST activity in the subcutaneous adipose tissue.

We thank for the reviewer for pointing out the error in the text. This mistake has been fixed and the corrected sentence is now found on lines 442- 443.

4. Line 540. Oscillibacter, Bilophila and Ruminiclostridium 5, which were significantly decreased with XOS
Please check the Ruminiclostridium 5.

The genus name is now stylized correctly (line 516).

Reviewer 3 Report

In this study, the authors evaluated whether xylo-oligosaccharides (XOS) affects adipose tissue inflammation and insulin signaling, and whether the gut microbiota (GM) and fecal metabolome explain associated patterns. They found that biclustering provides a useful algorithmic method for capturing such joint signatures. On the HFD, XOS-supplemented rats showed lower number of adipose tissue crown-like structures, increased phosphorylation of AKT in liver and adipose tissue as well as lower expression of hepatic miRNAs. XOS-supplemented rats had more fecal glycine and less hypoxanthine, isovalerate, branched chain amino acids and aromatic amino acids. Several bacterial genera were associated with the metabolic signatures. Although data provided here by the authors could have interesting implications, there are several concerns and problems in research design and experimental data. Specific points to be considered are listed below:

  1. For HFD and LFD animal model studies, the author should first provide body weight, liver weight and liver to body weight ratio.
  2. The experiment lacks mechanism study. Although studies have demonstrated that XOS prevented HFD-induced hepatic steatosis is associated with GM and fecal metabolome, the mechanism behind these effects is still unclear.

Author Response

Review points and responses

  1. For HFD and LFD animal model studies, the author should first provide body weight, liver weight and liver to body weight ratio.

We thank the reviewer for this relevant suggestion. On lines 124-126 we refer the reader to a supplementary table, which provides these requested parameters as well as effects of dietary fat and XOS on them. As body and liver weights are not further discussed in the text, we believe a supplementary table will provide the necessary information without interrupting the main text.

2. The experiment lacks mechanism study. Although studies have demonstrated that XOS prevented HFD-induced hepatic steatosis is associated with GM and fecal metabolome, the mechanism behind these effects is still unclear.

We have considered this valuable feedback and made several semantic corrections to the manuscript. We do not believe that mechanistic studies are possible in the scope of prebiotics, thus, we agree that our study has been largely associative. However, we also identified that, through the introduction and the presentation of results, we might have inferred something on the contrary. Key corrections in the abstract (line 15) and the intro (line 92) should provide a more realistic premise to the study. In the discussion, we have emphasized the associative nature of the study (lines 571, 632, 687, 702, 726-727) and the need for future mechanistic studies (lines 613-615, 629-630, 658-659). This is underlined in the conclusion section as well (lines 742-743). We hope that these changes satisfy the reviewer.

Round 2

Reviewer 3 Report

The authors have carefully addressed the reviewer's comments and improved the quality of the manuscript. I have no further comments.